# Plant Conservation vs. Folk Traditions: The Case of *Ophrys scolopax* Cav. (Orchidaceae) in Central Western Spain

**DOI:** 10.3390/biology11111566

**Published:** 2022-10-25

**Authors:** José A. González, Sonia Bernardos, Francisco Amich

**Affiliations:** Grupo de Investigación de Recursos Etnobiológicos del Duero-Douro (GRIRED), Facultad de Biología, Universidad de Salamanca, E-37071 Salamanca, Spain

**Keywords:** woodcock bee orchid, popular festivals, religious calendar, biocultural heritage, conservation

## Abstract

**Simple Summary:**

In Europe, numerous sacred natural sites act as a refuge for endemic, threatened, rare or specialist plant species, and they maintain a higher plant species richness, and have contributed significantly to vegetation structure and diversity at the landscape scale. However, in certain cases, inadequate planning and management of celebrations, ceremonies and ritual practices carried out periodically has led to a negative influence on the plant richness of these sites. In this article, we present an example of this and how it is possible to maintain the celebration of a popular festival while at the same time respecting and preserving a certain plant species. Furthermore, our study project shows the great importance of the participation and close collaboration of European rural communities in the conservation of biocultural diversity.

**Abstract:**

In central western Spain, the bee orchid *Ophrys scolopax* Cav. is limited to a few localities of the Arribes del Duero Natural Park, reaching the municipality of Villarino de los Aires (Salamanca) to the north. Due to its restricted distribution, this plant is hardly known in this territory, with the exception of this village, where it is very popular. Although most of its inhabitants are unaware of various aspects of the biology of this orchid, for example its pollination strategy, the place where the only local population grows is well-known: the Teso de San Cristóbal (“St. Christopher’s Hill”), a place of ancient pagan rituals Christianised through the construction of a hermitage. The villagers also know that its flowering period coincides with the Easter celebration, and they have traditionally looked for and collected it there during Easter Monday. This ritual has evolved over time based on the needs and interests of the community. From a religious celebration aimed at blessing the fields, it became a game among young men to obtain prestige within the community, and from the end of the 20th century to the present, it has become a festival to revitalise cultural identity. In this article, we analyse how the aforementioned traditional practices affected this orchid species in the recent past, and we describe the educational actions (conferences, workshops, courses, etc.) carried out during the last ten years so that, while maintaining the cultural practices of the village, its population should be respected and conserved at the same time.

## 1. Introduction

The interest in biological heritage often goes beyond the scientific and natural sphere and is often closely related to the historical and artistic heritage, to the beliefs, traditions and folklore of a given region or place, and may even have an important religious significance or become a sign of local identity.

There are places in which biological diversity conservation has been practised for many centuries in a community-based form, enclaves where religious beliefs, traditions and ethics, as well as the religious calendar, have traditionally marked the management regime, often imposing limitations on certain activities, so as to secure important resources and services for the whole community [1,2,3,4]. These are the so-called sacred natural sites (SNSs) that have been associated with a wide range of faiths and beliefs, socio-cultural systems, traditions and ritual practices, and may be subject to changing conditions [5,6,7].

Regarded as the oldest form of protection of habitats and other types of natural resources in human history [7], SNSs not only reflect the religious and social needs of the community, but at the same time contribute important ecosystem services, for example the conservation of biological diversity, among others [7,8,9,10].

While the sizes of SNSs around the globe vary greatly, many of them only covering an area of land of a few square metres, and conservationists regard the often small size of SNSs as a factor limiting their conservation value, there is strong evidence that even small-sized SNSs have a positive effect and a considerable conservation relevance in the case of plants [11,12,13].

For centuries, in Europe SNSs have been of great importance as plant diversity refuges—SNSs sometimes act as refuges for endemic, threatened, rare or specialist species [7], and they have maintained a higher plant species richness, and have contributed significantly to vegetation structure and diversity at the landscape scale [7,14,15], but the changes in the mentality and religious values of the new generations [16,17,18] have led to the conversion of the organisation of the main celebrations (in a significant number of cases) to a negative influence on the plant richness of these sites, where ceremonies and rituals must therefore require intense planning [14,19].

In this work, we illustrate how a link between religious, cultural, geological and biological values can be fundamental in the management, conservation and projection of biocultural diversity [20].

We present a study and a participatory conservation project carried out in an SNS where a very popular festival has negatively influenced the local population of a rare plant species. This is a clear example of the great importance of the participation and close collaboration of the European rural communities—who still retain complex conglomerates of traditional/local knowledge, practices and beliefs [20]—in the conservation of biocultural diversity. This project is also an example of how great successes in the conservation of biocultural heritage can be achieved without significant financial effort, simply through the active collaboration of the inhabitants of the area and the always invaluable help of colleagues from different disciplines (ethnobiologists, botanists, anthropologists, ethnographers, etc.).

### 1.1. A Sacred Natural Site

The study and participatory conservation project of biocultural heritage which we present in this article was carried out in the extreme northwest of the province of Salamanca, in the Arribes del Duero Natural Park (henceforth the ARD). This protected area extends along the banks of the river Duero and some of its main tributaries (the rivers Tormes, Águeda, Huebra and Uces) from the Spanish–Portuguese border (Figure 1).

The morphological homogeneity of the surrounding peniplain, the altitude of which ranges between 700 and 800 m a.s.l., is broken up by the gorges of the region. The river Duero and its tributaries run through an extraordinary labyrinth of canyons and gorges—*arribes*—with rocky cliffs that often exceed 400 m in height [21].

The area’s geographical and biogeographical location on the border between the Mediterranean and Euro-Siberian worlds makes it biologically rich and complex [22,23]. The floristic catalogue of the ARD includes 1105 species of vascular plants corresponding to 118 botanical families, and the main vegetation types include oak, holm oak and juniper forests, and scrub and riparian vegetation [24].

In this spectacular landscape, one of the most notable points is the hill known as Teso de San Cristóbal (“St. Christopher’s Hill”), henceforth the TSC, located in the municipality of Villarino de los Aires and within the granitic-migmatitic unit of the Tormes Dome [25]. It is a prominent granite hill in the form of a sill above migmatites, but the geological interest of this area is not only based on its structural aspect but also on its geomorphology. The TSC stands out because of the selective embankment of the river network [25]: with steep slopes on the granitic contacts and weak slopes on the metamorphic terrains, where remnants of old agricultural terraces remain, characteristic of the ARD due to the mesoclimatic conditions [26], and where gentle valleys can be observed (Figure 2).

For its very high geological value (which makes it stand out from the surrounding environment) and for its scientific interest, the granitic sill of the TSC was included, together with another interesting geological formation nearby, in the Spanish Inventory of Sites of Geological Interest—IELIG, for the Spanish name Inventario Español de los Lugares de Interés Geológico [25]. This is an inventory of the Spanish geological heritage, the main purpose of which is to promote the sustainable development of “geosites” and to leave a well-preserved environment for future generations [27].

However, beyond its biological and geological interest, the TSC is closely related to the cultural heritage treasured by the community of Villarino de los Aires, with its traditions, folklore and popular religiosity. The TSC, markedly flat on its summit and inhabited since Neolithic times, is a place of ancient pagan rites and cults Christianised through the construction of a hermitage [28,29,30,31,32,33]. The constantly enigmatic presence of cazoletas (bowl depressions in the granite stones) and a dolmen facing south (related to the cult of the sun), as well as the medieval anthropomorphic tombs in the vicinity of the hermitage, testify to the special significance and symbolism of this rock sanctuary [33,34].

Found in an outstanding geomorphological location and existing for many centuries or even millennia, as other important European SNSs, the TSC has acted as a refuge for threatened, rare and specialist plant species, but the main festival held annually on this hill has had a negative influence on the plant richness of this site.

### 1.2. A Popular Festival

Throughout history, the TSC has been a sanctuary to which the inhabitants of Villarino de los Aires have periodically come to worship on important dates [32] to their corresponding deities—assimilated to the rivers [34].

According to data obtained by direct consultation of two accounting books (from 1626 and 1737, respectively) kept in the Archives of the Diocese of Salamanca, in the 17th and 18th centuries religious processions were held from the village church to the TSC and masses were celebrated in the hermitage there for the blessing of the fields and to avoid bad storms.

As can be concluded from the study of these two ancient texts, the origin of the festival we have studied and will describe was to bring about good harvests, a real necessity for an agricultural community dependent on climatic processes. However, the rites celebrated on the TSC have changed over time, as well as their meaning. The procession to honour the pagan gods became a Christian pilgrimage, and a certain swaying rock historically used as a sacrificial throne and altar and mentioned in the first of the two books (“*the highest peak of the hill, and although it is so large, any man of skill can move it*”), became the pedestal for the Christian ensign of a brotherhood [32,33].

As anthropologist David Ch. Tilley Bilbao [35] postulates, the ritual has been (and is) articulated as a malleable mechanism of support for the community, which has used (and uses) it according to its own needs and interests, which evolve over time. Thus, the event popularly known as the Fiesta del Teso (i.e., “Festival of the Hill”) has as its main facet of cultural interest the placement of a banner on the highest point of the hill, in recent times replaced by simply placing a plant element (usually a branch of a tree).

In 1905, the writer and philosopher Miguel de Unamuno recounted his impressions and experiences in a visit he made in 1902 together with other scholars to the ARD (Figure 3a,b). He wrote [36]: “*Before entering Villarino, [...] we took a detour to go up to the Teso de San Cristóbal, where a pilgrimage was being held that day, the 1st or 2nd of May. [...] On that hill […] in front of the hermitage, on an area of level ground, boys and girls were dancing, […] A huge granite rock, almost circular, crowned by a small flag, presided over the celebration*”.

In the 1940s, the Augustinian priest César Morán, a member among others of the Sociedad Española de Antropología, Etnografía y Prehistoria (Spanish Society of Anthropology, Ethnography and Prehistory), wrote [29,37]: “*On the day of the festival, once a year, all the people of Villarino parades there, they hear mass, eat, drink, dance and the young men place the flag on top of the swaying rock*”. These two authors mention the placing of a flag by the young men of the village on the top of the large oscillating tor located on the highest point of the TSC, known as the Peña del Pendón (i.e., “Rock of the Banner”) (Figure 3), but they also note the courtship dances performed by the men and women.

Nowadays, with the restoration of the hermitage at the end of the 2000s, as well as having recovered the mass in honour of St. Christopher (on 25th July), mass is still celebrated on Easter Monday and, afterwards, people eat hornazo (a typical local meat and pork sausage pastry), drink and dance [32,38]. The country picnic aspect of this festival dates back a long way. In what is known as the Catastro de Ensenada, a census of inhabitants, territorial properties, buildings, livestock, trades and income ordered by King Ferdinand VI in 1749 to his minister Zenón de Somodevilla, Marquis de la Ensenada, apart from other details, the following expenditure is recorded [32]: “*On the feast day of St. Philip, refreshments are given in the chapel of San Cristóbal to the villagers who go on pilgrimage*”. Also known as the Día del Hornazo (i.e., “Hornazo Day”), this festival is attended by all the people of the village, as well as other people from neighbouring communities, and the tradition of placing a branch of a tree or bush on the top of the Peña del Pendón still continues (Figure 3).

This great stone is, possibly, the most interesting sacred swaying rock in all of western Spain [29,30,39]. Considered to be a rock altar by some authors and related by others to the “rock thrones” because of its shape like a seat, the local tradition associates the capacity to make it sway with an ordeal, as it is located on a precipice from which it is said that the damned were thrown, an interesting fact for its ritual assessment [33,39].

During the last century, the hanging of the banner still consisted of a game of prestige among the young men of the village, obtaining it within the community was one of the few opportunities to acquire it: “they would arrive as early as possible, put up the banner and boast about it for the whole year”. Nowadays, this tradition has been relatively abandoned, but there is also a process of revitalisation of the ritual by some of the inhabitants: “Last year I went at dawn to put up the banner (a holm oak branch, as long as possible), and when I reached the TSC, to my surprise someone had beaten me to it, it filled me with pride that someone else is continuing the tradition” [35].

In 1937, Antonio García Boiza, Professor in the Faculty of Philosophy and Arts at the University of Salamanca, wrote [28]: “*The ruins of a temple in the Teso de San Cristóbal, on a beautiful hill from where you can enjoy magnificent scenery. The most varied flowers and irises grow on this hill, as well as a flower, the “avispina”* (the woodcock bee orchid)*, which can only be found here. They celebrate the pilgrimage, one of the most typical in the region, on the 3rd of May*”.

This document, written at a time when the hermitage was in ruins (the original hermitage was collapsed for much of the 20th century) and when religious celebration at the end of worship was of little relevance to it [33], indicates to us the survival of the festivity, at least in its most playful aspect, but it also points to the existence of a great botanical richness.

The TSC has been proposed as a Plant Micro-Reserve, a protection status established in the Catalogue of Protected Flora of the Autonomous Community of Castile and Leon—Decree 63/2007, BOCYL No. 119, of 20 June 2007 (https://medioambiente.jcyl.es/, accessed on 13 July 2022)—with the aim of protecting small areas of maximum interest for being the site of outstanding populations of the most endangered plant species. Like other sites in Castile and Leon [40], the TSC might be subject to threats related to recreational activities, especially during late spring–early summer, coinciding with the blooming periods of several rare, endemic, threatened and of interest species. As such, *Delphinium fissum* subsp. *sordidum* (Cuatrec.) Amich et al., listed in the Red List of Spanish Vascular Flora [41] as Endangered (EN) and in the Catalogue of Protected Flora in Castile and Leon as “In danger of extinction”, presents the highest density of reproductive individuals in the TSC [42].

The Día del Hornazo coincides with the flowering season of numerous irises (*Iris* × *germanica* L., Iridaceae) which grow over much of the TSC, and here we come to describe the botanically interesting aspect of this festival, as young men included these beautiful flowers in their courtship ritual.

The young men would gather irises and make bouquets of them to offer to the woman they loved, or the one they wished to win. The iris is a flower loaded with symbolism, being considered a pure and divine flower that evokes the high dignity and the chivalry of royalty or the papacy [43,44]. Its religious significance is linked to the Virgin Mary: it is a symbol of purity, innocence and virginity, of the virginal conception of Jesus and of the fact that Mary was born without original sin [45,46,47]. In Christianity, the iris became the symbol of pure and virginal love [48]. To reinforce its value as an offering of love, we think that, at some point in history, the young men found among the irises such strange, curious and striking flowers that they incorporated them into their bouquets—it was the woodcock bee orchid, *Ophrys scolopax* Cav., the “*avispina*”.

### 1.3. An Emblematic Plant Species

*Ophrys* L. is the genus with the highest number of species of all the Mediterranean and European genera of Orchidaceae. *Ophrys* is distributed from the Canary Islands to the Caspian Sea, and from the south of the Scandinavian Peninsula to North Africa, and it is especially common in the Mediterranean Basin and in the radiating regions of its flora and plant communities [49]. Some of the areas in which *Ophrys* shows a very high diversity are the Aegean region, southern Italy, the Maghreb and the Iberian Peninsula. Nelson [50] hypothesises that the eastern Mediterranean would be its centre of origin, given the very high diversity of species in these eastern territories, although other authors point out that, at least the section *Pseudophrys* would have had its origin in Mediterranean Africa [51].

The number of species considered to be in the genus *Ophrys* is very variable according to the different authors consulted. In the Iberian Peninsula, for example, between 36 [52] and 12 species [53] are indicated. Different positions and arguments regarding nomenclatural and taxonomic characters cause these differences in the number of species considered.

*Ophrys scolopax* (Figure 4) is distributed throughout the western Mediterranean Basin, in particular the Iberian Peninsula, southern France, Liguria and the Maghreb [52]. Some references from the centre of the Basin have not been confirmed [54]. In the Iberian Peninsula, it seems to be scattered throughout most of the territory and it is widely represented in those areas where basic rocks predominate [53]. However, these clear preferences for basic substrates make it rare and scarce in areas characterised by acid rocks—granites, quartzites, schists and slates. Therefore, in the central-western area of the Iberian Peninsula, it is almost exclusively limited to a few localities of the ARD in its part of the province of Salamanca [55,56], reaching as far as La Fregeneda in the south and Villarino de los Aires in the north (Figure 1).

Among the seven known populations of this plant species in the territory of the ARD, the northernmost one stands out, precisely the one that develops in the TSC (Figure 1 and Figure 2), the threatened and conservation target population. In this place of undeniable historical and cultural importance, the species preferentially integrates into more or less humid grasslands, on acid rocks, generally granite. It is a scarce plant, irregularly distributed across the hill, and during the years of study no more than 20–25 specimens were counted.

Generally speaking, in the other six localities this plant is not known by the inhabitants; however, in Villarino de los Aires, where it is called *avispina* (i.e., “little wasp”) because of its wasp-like flower, it is really very popular. This vernacular name is unique in the Iberian Peninsula. In his *Estudio sobre el Habla de La Ribera* (1947), Llorente Maldonado wrote [57]: *Avispina*: “Gramineceous plant parasitic on the iris bulb, similar in shape and colour to a small wasp and which has only been found in the Sierra Nevada and the Villarino hillsides”. The only species it could be confused with is *O. apifera* Hudson. However, this bee orchid species is even scarcer and rarer than *O. scolopax* in the ARD territory, and has not been recorded in our study area, Villarino de los Aires.

On the other hand, most of the inhabitants are unaware of its pollination strategy, pseudocopulation (sexual deception) with male bees of the genus *Eucera* Scopoli, 1770 (Hymenoptera, Apidae), mainly [52,58], or they are simply unaware that it is an orchid, if the place where a given population lives is traditionally known: the TSC. Additionally, the villagers know that its flowering coincides with the celebration of Easter—this bee orchid blooms in early spring, from late March to late April [53]. Traditionally, this plant has been looked-for and collected on Easter Monday, a very popular local holiday, as mentioned above.

For all these reasons, this orchid is deeply rooted in the collective memory and very present in the mentality of the inhabitants of Villarino de los Aires. This plant is considered as a symbol of identity, as something representative of its community, a rural Spanish community where a clear relationship has been built up between popular religiosity, magic and the flora [59].

In this work, we analyse the level of knowledge of the inhabitants of Villarino de los Aires in relation to the plant species studied, evaluate how the popular festival celebrated in the place where its only known population occurs in the territory of this municipality (the most north-western population of its known global distribution) has affected this bee orchid in the past and currently threatens it, and we describe the actions carried out to maintain and conserve this population as part of the biocultural heritage of future generations in this part of central-western Spain (the main objective of our effort).

## 2. Materials and Methods

As an initial stage of the biocultural heritage conservation study project presented here, between the months of February and March 2011, structured interviews were conducted with a total of 242 people, more than 25% of the 950 people registered in Villarino de los Aires at that time. Prior to each interview, verbal consent was obtained from each of the participants and the Code of Ethics adopted by the International Society of Ethnobiology [60] was followed.

A questionnaire was compiled with the aim of gaining insight into the attributed values and associated symbolisms to *Ophrys scolopax* in this Spanish rural community. The questionnaire had various items referring to people’s knowledge about the biology and phenology of this species (Appendix A).

A stratified random sampling by gender and age was employed. Thus, the questionnaire was given to persons of both sexes (121 men and 121 women) and of all ages (121 under 60 years and 121 aged 60 years or older). Children under 6 years of age were not included in the study (Appendix A). Each person interviewed was asked their age, place of birth (Villarino de los Aires vs. other location) and level of studies (primary education vs. higher education).

With all the data obtained from the various interviews conducted in Villarino de los Aires, a database was constructed using Apache OpenOffice (formerly OpenOffice.org) software v4.1.13. To analyse the independence or otherwise between the qualitative variables considered in relation to the characteristics of the different informants (gender, place of birth and educational status) and the collection of specimens of the plant studied, we performed different two-by-two contingency tables and the corresponding chi-squared statistical tests (significance level α = 0.05). In the case of the age variable, although it is quantitative and for each informant we noted the number of years completed at the time of the interview, taking as a basis the results of the analysis of informants’ knowledge in previous studies carried out in the ARD area [59,61,62], we chose to group them into two categories: “young” informants (with little knowledge), aged less than 60 years, and “older” informants (with a sound knowledge of useful plants), aged 60 years or more (in 2011, i.e., born before 1952). Below the threshold of 50 years of age, the TK amassed by the different informants is insignificant.

In addition, the data related to the collection of specimens by the 242 people interviewed were represented in a chord diagram generated with the free online generator of Datasmith.org (http://www.datasmith.org/2018/06/02/a-bold-chord-diagram-generator/, accessed on 10 September 2022).

On the other hand, to confirm that woodcock bee orchid is unknown in the other localities of the ARD where populations occur, a series of interviews was also conducted. In this case, 10 informants (5 men and 5 women) over 65 years of age and with a high level of flora-related TK were chosen from each village, and as a starting point for these interviews we presented each informant with a printed photomontage including three photographs. To avoid recognition of the plant from the environment, we avoided presenting photographs of local specimens. Therefore, the following were included: a photograph of a specimen from another population of the ARD and two of the images included in the Spanish Wikipedia (http://es.wikipedia.org/wiki/Ophrys_scolopax, accessed on 21 May 2011).

Following the line of Clifford Geertz [63], in that explanations for cultural facts correspond to the people who create/elaborate them, in 2012 semi-structured interviews were conducted with a focus group of key informants, as a way of approaching the reality of the circumstances and achieving a direct analysis of symbolic systems in Villarino de los Aires.

## 3. Results and Discussion

### 3.1. First Step: Analysing How the Aforementioned Traditional Practices Affected the Studied Orchid Species in the Recent Past

#### 3.1.1. Who Knew of This Orchid?

In those other villages of the ARD where populations of *Ophrys scolopax* grow, only seven of the people interviewed recognised the plant in the photographs shown to them. In the neighbouring village of Pereña de la Ribera, two women aged 70 and 66, housewives, and a retired man, aged 80, told us that they knew the plant from Villarino de los Aires, “where they call it *avispina*”, that they were going to picnic on Easter Monday at the TSC and that in their municipality “there is no such plant”. In Masueco, while a 67-year-old woman told us: “I know it from Villarino de los Aires, from the TSC, there they call it *avispina* (we go nearly every year to picnic there on Easter Monday)”, a retired 89-year-old shepherd said: “I remember having seen, as a child, a flower like these, which looked like a bee (it was very pretty). It was in the area of the Uces river canyon (there are many different plants there). Here, in Masueco, this plant has no name”. In an almost identical way, in Vilvestre and La Fregeneda, the most distant localities from Villarino de los Aires, two other retired shepherds (78 and 76 years old, respectively), told us: “I remember being with the sheep one day, in early spring, and seeing this plant. It caught my attention, the flower looks like a bee, it is of the same shape and the same size. I don’t remember having seen it any other time”. “I have seen it a few times on the way down to the Duero River, in the place we call Alabancos. The flower looks like a little bee”.

These data show that, for flora-related TK, *Ophrys scolopax* is also a rare and scarce plant species in the territory of the ARD, and that many of the people who know it, even if they are from another locality, associate this curious orchid with Villarino de los Aires and the TSC.

In Villarino de los Aires, only 23 of the 242 people interviewed (9.5%) did not know of this orchid. However, we noted that all these people had heard of it from their elders.

There was a greater lack of knowledge among the children under 10 years of age. Only 18% of the children interviewed recognised the plant.

Among the inhabitants aged 10 to 29 years of age, 76% knew it. The informants in this age group associate the Easter Monday festivity with a day of picnic with the whole family and friends, with the music of a charanga, and the custom of two heifers fighting, which for some years took place (not nowadays) in the rustic bullring built for this purpose in the TSC (Figure 3g).

The few older people who said that they did not know the plant, said that this was due to “personal matters”, the impossibility of getting to the TSC on the day of the festival because of hard work in the fields, widowhood, etc.

#### 3.1.2. How and When Did you Know of the Plant?

A total of 94 of the 219 people (43%) who knew the plant indicated that it was members of their family who first taught them about it in the TSC. In particular, they cited their parents as being most important in transmitting this TK. Another 49 people (22%) indicated that it was their friends who showed it to them in situ, and 19 (9%) mentioned their neighbours—who showed them the plant in the TSC in some cases, but mostly transplanted in a pot.

It is also interesting to note that 51 people (23%) acknowledged having seen the plant for the first time in photographs. They mentioned photographs included in different media and formats. In the book announcing the patron saint festivities of St. Roch (16 August), a book that was published for several years in the 2000s and in which photographs of the *avispina* were included, among others relating to the architectural and landscape values treasured by Villarino de los Aires. In the Municipal Tourist Office, promotional photographs of the village can be seen, among which this orchid also appears, and until 2009 there was Villarino TV, a local TV channel on which its image was included in numerous documentary reports and, especially, in promotional advertisements for the Día del Hornazo festival. It should also be noted at this point that many people have a framed photograph in their homes of *O. scolopax*, which they enjoy showing to their neighbours and visitors.

Finally, the six people remaining knew of the plant one day when searching for it by themselves, guided only by the description of it heard throughout their lives from numerous neighbours and relatives.

#### 3.1.3. What Did They Know about the Plant?

All the villagers interviewed who knew the plant species (219 people) gave it the name *avispina*: 199 of them said they named the plant after its flower, which resembles a wasp—*avispa* in Spanish. Interestingly, 45 people said that the flower actually looks more like a bee than a wasp.

Only 68 respondents (31%) knew that it is an orchid, and only 24 (11%) knew about the pollination process of this type of orchid. Of the remaining people interviewed, some dared to postulate that the curious shape of the flower serves the plant “to camouflage itself”, “to defend itself and scare away predators”.

Regarding the area where this plant species grows, 213 respondents (97%) mentioned the TSC as the only place in the municipality where it can be found. Only six people mentioned having seen it elsewhere at some time (remember that in Villarino de los Aires the only population known to botanists is that of the TSC). In relation to the flowering season, all the respondents who knew the plant mentioned spring, especially the first days of May and coinciding with Easter Week, and more specifically with the Easter Monday festival.

#### 3.1.4. How Many People Have Collected It? When and How Did They Collect It?

Of the respondents, 62 (29 men and 33 women) claimed to have collected this orchid at least once in their lives, representing just over 25% of the total number of those interviewed.

Another 31 people (50%) collected it on the day of the festival, the Día del Hornazo, while the other half said they had collected it on another day (“before the day of the festival”, “when there was no one around”, etc.). One informant even mentioned collecting it at night.

As for the way of collecting it, 27 people (43.5%) said they had collected it “with the potato”, i.e., by digging with any simple tool and pulling out the tubers. The remaining 35 people (56.5%) preferred at the time to simply cut the “flowering stem”, collecting its flowered aerial part.

#### 3.1.5. Who Collected It? (A Robot Portrait)

The collection of specimens or samples of this species is independent of gender. There are no significant differences between men and women (χ^2^ = 0.3469, df = 1, *p*-value > 0.05), and this variable is not related to the way in which they collect the plant: with the tubers or only the flowered aerial part (χ^2^ = 0.1043, df = 1, *p*-value > 0.05).

There is an association between the origin of the people (those born in the village) and the collection of specimens of the species (χ^2^ = 25.0216, df = 1, *p*-value = 0.000). Thus, a total of 49 people born in Villarino de los Aires have collected this orchid (79% of all those who have collected it), to which can be added the 6 people interviewed who were born in the hospital in the city of Salamanca and have resided in the village throughout their lives, which brings the proportion to 89%.

Similarly, the collection of the plant is not independent of age. Therefore, grouping those interviewed into two categories: “young”, aged less than 60 years, and “older”, aged 60 years or more, the orchid was collected especially by people in the latter category (χ^2^ = 17.0008, df = 1, *p*-value = 0.000); specifically, 45 of the 121 people interviewed aged 60 to 99 years (37%). It should also be noted that more than half of those interviewed aged 70 to 79 years (54%) collected it.

Finally, another qualitative variable analysed was the educational level of the informants, the educational status. Most of the people who admitted having collected the plant (79%) had only primary education (χ^2^ = 5.8519, df = 1, *p*-value < 0.05), in some cases not comparable to formal education. In this case, the result is not so markedly significant and, although it allows us to reject the null hypothesis, it indicates to us that the variables “age” and “place of birth” are those that should have a greater weight in the planning and development of the conservation project. People aged 60 or over and born in Villarino de los Aires will be a key part of the project (Figure 5).

#### 3.1.6. Why Did They Collect It?

Most of the people who collected this plant at some time (64.5%) indicated that they did so because of its rarity and, in particular, for its aesthetics, for its particular beauty (“it is very pretty”, many emphasised). Some 21% said they had collected it because it was “a custom of the village”, “it is something from the village”. The remaining 14.5% (nine people) attributed to this strange flower the property of being a powerful talisman (“it brings good luck to whoever possesses it”). A 70-year-old woman recalled how her mother kept a dry *avispina* among the leaves of her missal.

#### 3.1.7. What Did They Do with the Collected Specimens?

In relation to the last comment, seven of those interviewed said that they dried and kept the gathered plant (its flowered aerial part) between the pages of a book. “Put it in water”, in a vase, was the answer of 29 respondents to the corresponding questionnaire item, and 24 of the 27 people who claimed to have ever collected it with its tubers transplanted it into a pot. In all cases, they claimed to have been unsuccessful in their cultivation.

From an anthropological point of view, we can observe how the *avispina* flower has become an identifying symbol of the inhabitants of Villarino de los Aires, in which the significance of the symbol (perception and interpretation between the signifier and the meaning of the symbol itself) has changed over time.

Through the interviews carried out in the fieldwork, we can observe (and confirm) how, prior to the 1950s, a game of prestige was established around the woodcock bee orchid.

During the fiesta held at the TSC, single men competed with each other for the flowers in order to give them to the single women they were interested in. It was a competition in which single men gained prestige. In the words of one of those interviewed: “The one who didn’t manage to catch that flower was the dumbest in the group” (excerpt from an interview conducted on 17 March 2012).

The men competed to catch one of the orchids and give it to the single woman in whom they were interested. The women who accepted the flowers would display them in their hair or in their clothing; in this way, the flower became an item to make the young men’s interest in a romantic relationship visible.

From this date, and especially during the 1960s, there was a lot of immigration among the people to large urban centres in Spain (Madrid, Barcelona, Bilbao) and to other countries, mainly in Europe [64,65]. During this migration, the flower of the *avispina* is seen as an identifying element of the village. Several of the immigrants try to take the plant with them as a souvenir of the village. We can see how there is a correlation between the plant, considered as something unique to the community, and the village of Villarino de los Aires itself.

In this way, the *avispina* constitutes an identifying symbol, in which the signifier is the flower and the signified is the village of Villarino de los Aires itself. The immigrants, on taking the flower, carry a symbol of its place of origin. As one person said in an interview: “It is as if they are taking a part of the village with them” (fragment of an interview conducted on 19 March 2012).

### 3.2. A Plant Conservation Educational Project

Apart from being clearly affected by factors such as habitat modification and climate change, orchids are very vulnerable to human actions, in particular they are globally threatened by unsustainable harvest for horticulture, food or medicine, illegal collection and tourism and recreational activities [66,67,68]. Therefore, taking into account that each plant in a population is irreplaceable, that when a population disappears it is very difficult to recover and it is very complicated and costly to reproduce orchids ex situ, in order to avoid the reduction of the local population of *Ophrys scolopax* and the impoverishment of its genetic diversity, which increases the risk of disappearance, various educational actions (conferences, workshops, courses, etc.) have been carried out in Villarino de los Aires during the last 10 years. These activities have always been directed towards maintaining the cultural practices of the village, while the local population of the woodcock bee orchid is respected and conserved at the same time.

#### 3.2.1. Educational Actions Developed

Several workshops have been held and talks conducted and an introductory course on botany has been given to make the community of Villarino de los Aires aware of the enormous wealth of flora that exists in the ARD, which in this protected area 28 species of orchid are known [56], and that *Ophrys scolopax*, known locally as *avispina*, does not grow only in the TSC (as was believed in the past), but there are other populations, both in the territory of this Natural Park and elsewhere in the Iberian Peninsula.

Conferences have also been held on the special reproductive biology of this type of orchid, plants that have very precise mechanisms of pollination and germination of their seeds. Bee orchids have elaborate systems for attracting pollinators, restricting their distribution to that of their specific pollinators [69].

In the genus *Ophrys,* one speaks of sexual deception or deceptive pollination. Stimuli of the types visual (morphology), tactile (hairiness) and olfactory (female sex pheromones) are used by these curious orchids to attract pollinators, and a high specificity of attraction of a male bee followed by complex pseudocopulation behaviour to correct pollen transfer within the species is observed [70,71,72,73,74]. This specificity is caused by a highly remarkable imitation of the sex pheromones of the mimicked female of the pollinator. Flowers mimic the odour of a receptive female to attract males that seek to copulate with the false female [74,75]. Different authors [76,77,78] have demonstrated that the *Ophrys* flower mimics the chemical compounds exactly like those produced by the true females of the pollinating males. Olfaction is used to orientate from a distance; however, in the near vicinity, where spatial vision is sufficient to detect the flower, the males’ approach is only visually guided [79,80]. Thus, to further ensure success, the pollinating males also select other characters of the *Ophrys* flower such as labellum size, colour, labellum hair characters, phenology and possibly habitat selections, etc. [58].

As already mentioned, *O. scolopax* is mainly pollinated by long-horned bees (*Eucera* spp.) [58] and, at this point, it has also been shown that the role of male insects in pollination has implications for the balance of the habitat. Bee orchids have complex and delicate interactions with their pollinators, which makes them particularly prone to local extinction. They are pollinated by specialised bees often occupying otherwise narrow ecological niches (e.g., pollen specialisation, a specific nesting site). This condition makes the orchid–pollinator interactions very fragile [81]. Solitary bees, one of the best documented and most studied insect groups, are experiencing decreases in their populations and ranges in Europe [82,83]. For example, between 1980 and 2013, in Britain, solitary bees suffered an average 32% reduction in range [84]. For many species, changes in landscape structure can be hugely detrimental as they often have specialised nesting and forage requirements.

Regarding the complexity of the process of seed germination in these orchids, it is essential to show the human community that this is due to the morphology of the seeds, nearly microscopic in size [85], and that they do not have any reserve substances and need the participation of a fungus to germinate.

The specific biological association of bee orchids with fungi for germination, growth and nutrients means that individual orchid species often occur as small, isolated populations, heightening their risk of extinction [86]. Bee orchids are mycoheterotrophic during their seedling stage and in many species the dependency on fungi as a carbohydrate source is prolonged into adulthood [87].

For all these reasons, it is safe to say (and it must be stated) that seed germination in this orchid group is significantly influenced by medium composition [88] and the plants require different growing conditions as compared to other normal flowering plants. As a result of this, bee orchid cultivation (i.e., pot culture, vegetative propagation, growing from seed) is quite difficult and it requires special management practices and special care [89,90]. It is also necessary to explain that ex situ conservation of endangered bee orchid species is one of the most important objectives in botanic gardens, but there are some difficulties in germination and growth of native threatened species [88,90]. Efforts to conserve this group of orchids demonstrate the difficulties in effective conservation [91]. Seed germination requires unique practices to propagate [90], and to conserve their genetic diversity it is necessary to develop a collection of seeds and orchid fungi [91].

#### 3.2.2. Impact and Social Implications of the Project

Having achieved the objective of reaching increased awareness and recognition of threats that place the local population of *O. scolopax* in Villarino de los Aires in danger, an improved management was sought to reduce human impacts. For this, a small list of recommendations was drawn up for the inhabitants of this village. Therefore, the importance of not collecting this plant was discussed, especially not pulling it up with its tubers. Due to factors such as small, dispersed populations, specific symbiosis with fungi and with pollinators and their desirability for collecting [66], this bee orchid is threatened by intentional collecting and by human intrusions and disturbance from recreational activities (Figure 6). For this reason, emphasis was also placed on making people aware not to step on it and not to disturb the environment.

A slogan was created: “Enjoy it without damaging it”, and the need to respect the existing signage in the TSC related to the conservation of the population of this plant species was emphasised—there is a series of metal signs prohibiting grazing and, literally, touching the plants. We have also managed to get small groups of people who, guided by us, go for walks to the TSC around the time the species is flowering to photograph it or take short films of it (Figure 7). The sensitivity of the people has been raised to preserve the population of this bee orchid that grows in their municipality, as well as to ensure the maintenance over time of this representative symbol of their popular culture.

However, all these tasks go beyond this, and the need to create initiatives for intergenerational dialogue that foster TK transmission has been considered in all phases of the project. With the children, actions were carried out in the field of Environmental Education, with the collaboration of the teachers from the Municipal School of Villarino de los Aires and/or older people from the village with extensive TK (Figure 7). These activities, like similar ones carried out in the field of Ethnobotany in different countries [92,93,94,95], were very satisfying for the children, but also for the adults.

Active ageing is a topic of particular relevance in Europe and in this area, TK is a valuable tool to promote intergenerational dialogue and, consequently, care for our elders. As Vallejo et al. [96] postulate, the meeting between generations can enhance the life experience of older people and the development of bidirectional competences as a result of this interaction.

On the other hand, we have also promoted different artistic works related to the beauty and conservation of this plant by people born or living in the village. In particular, we have worked to raise awareness among the inhabitants of the poems that some neighbours have dedicated to this plant [97] (p. 383) and the artistic work of Neil Allen, other valuable elements of the rich cultural heritage of Villarino de los Aires that were only known to 106 (44%) and 33 (14%) people, respectively, when interviewed.

Neil Allen is an English painter based in Villarino de los Aires who captured the beauty of this plant in his work entitled “Bee Orchid”, a painting which won him the prestigious BBC Wildlife Artist of the Year award in 2009 (www.neilallen.com, accessed on 5 July 2013).

Finally, we can declare that thanks to the effort carried out and the cooperation from local people, the population of *Ophrys scolopax* that grows in the TSC should currently be considered stable. The collection of specimens has been stopped and an increasing number of flowering individuals can be observed each year. This curious and beautiful plant will be enjoyed by future generations of the rural Spanish community studied.

## 4. Conclusions

Nowadays, and following the information and awareness campaigns carried out in the community of Villarino de los Aires (Salamanca, Spain) on the conservation and protection of the local population of the orchid *Ophrys scolopax*, this herbaceous plant is considered a rare asset to be protected. It is no longer simply a symbol of identity, but an element felt as belonging to this Spanish rural community which, like many others, is immersed in a process of loss of cultural identity.

Overriding traditional management can negatively affect a site like the TSC, where the vegetation diversity is indeed the outcome of both abiotic factors and active traditional management. We rather recommend traditional management to be encouraged at this SNS within the Arribes del Duero Natural Park, especially when this is supported by local people and traditional uses and cultural celebrations are still alive. This option would be the most effective way to conserve both flora and culture at the TSC, one of Europe’s last hotspots of biocultural diversity.

From a purely legal perspective, we propose the recognition of the TSC (found in an outstanding geomorphological location) as a hotspot of biocultural diversity, because by combining the biological and cultural heritage of the territory, its cataloguing should not merely be considered as a Plant Micro-Reserve.

## Figures and Tables

**Figure 1 biology-11-01566-f001:**
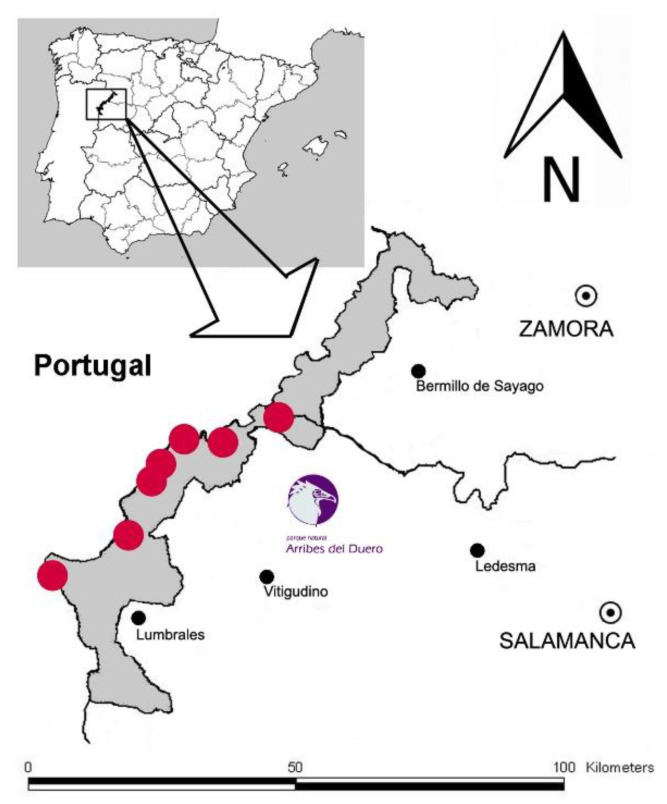
Geographic location of the Arribes del Duero Natural Park and the seven known populations in this protected area for the plant that is the subject of our study and conservation (red dots).

**Figure 2 biology-11-01566-f002:**
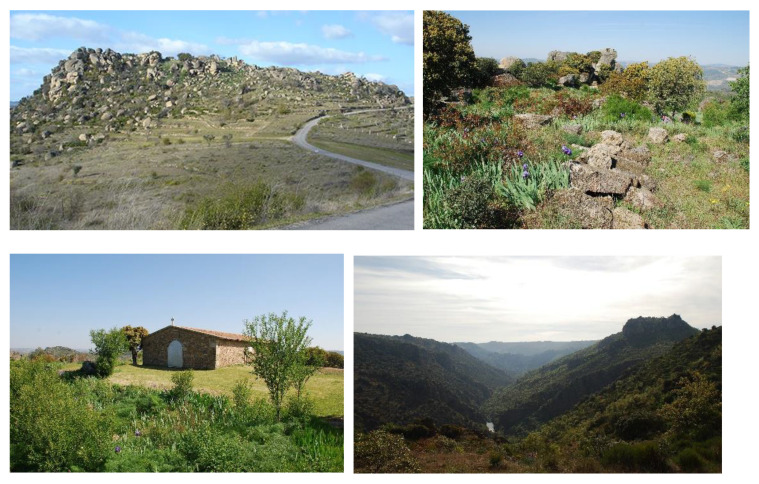
Images of the Teso de San Cristóbal, Villarino de los Aires, Salamanca, Spain (photos by J.A. González).

**Figure 3 biology-11-01566-f003:**
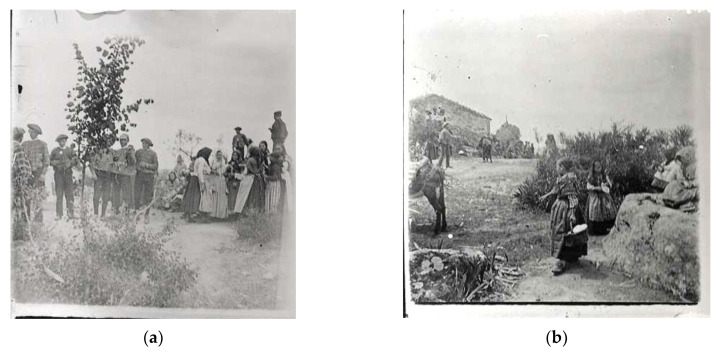
Different images of the celebration of the festival in the Teso de San Cristóbal (TSC) in recent history: (**a**,**b**) Photographs taken by Venancio Gombau in 1902 during Miguel de Unamuno’s visit to the TSC. (**c**) Around 1960, on the day of the festival, the young men adorned themselves with flowers on their hats (they even decorated their equines with flowers). (**d**) Blessing after mass at the hermitage. (**e**) At the end of the mass, the Town Council invites all the villagers to drink sangria, and eat olives and chochos (pickled lupini beans). (**f**) The younger villagers dance to the sound of the charanga band. (**g**) For some years, on this day of popular celebration, a couple of heifers fought in the bullring at the TSC. (**h**) Peña del Pendón, located on the highest point of the hill and facing east–west. On its east side, there are carved steps leading to the top, where there is a series of hollows (“thrones”) and a hole where traditionally the banner has been placed (photos by J.A. González).

**Figure 4 biology-11-01566-f004:**
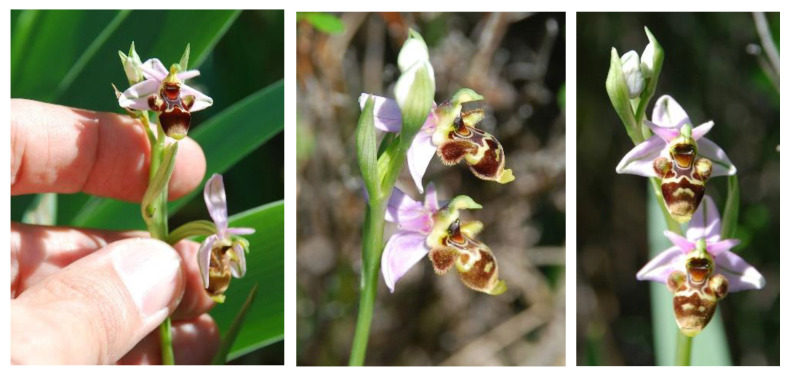
The studied plant species: *Ophrys scolopax* (woodcock bee orchid). Plant 10–40 cm tall, with inflorescence of up to 12 flowers. Sepals ovate-lanceolate, white to purple-violet, with a green midrib. Petals triangular-lanceolate, similar in colour to the sepals. Labellum 3-lobed at the base and reddish-brown to purple. Lateral lobes small and densely pilose, and the median with dense marginal pilosity. Basal field reddish-brown, surrounded by an H- or X-shaped speculum. Apical appendix distinctly conspicuous, acute, greenish yellow (photos by J.A. González and F. Amich).

**Figure 5 biology-11-01566-f005:**
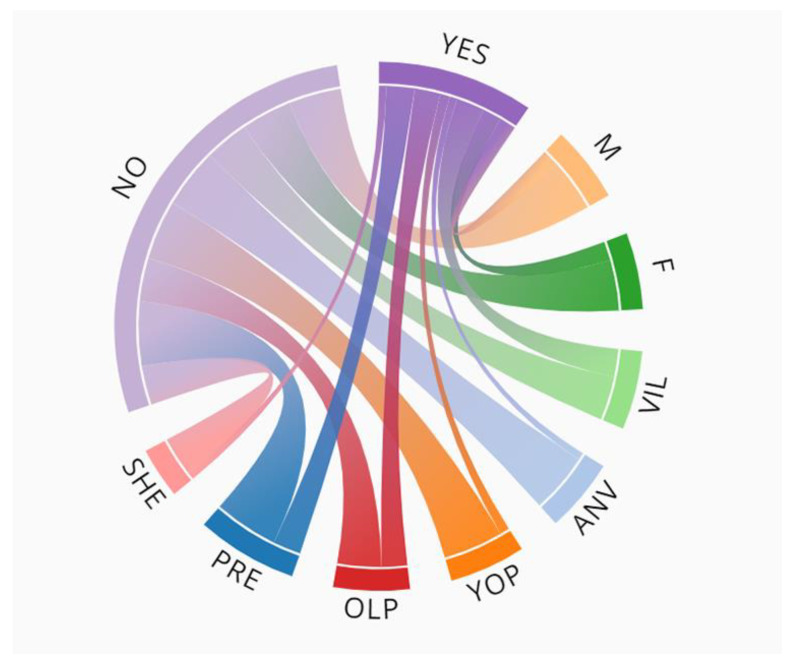
Chord diagram of collection of specimens and sociodemographic characteristics of the 242 people interviewed. M = male, F = female, VIL = born in Villarino de los Aires, ANV = born in another village, YOP = young people, OLP = older people, PRE = primary education, SHE = secondary (and higher) education.

**Figure 6 biology-11-01566-f006:**
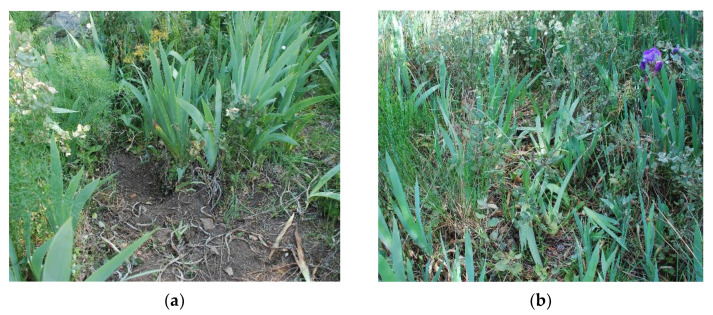
Examples of human impact in the environment of the Teso de San Cristóbal in the active search for specimens of *Ophrys scolopax*: (**a**) excavated and cleared soil by people looking for the tubers, and (**b**) trampling of basal leaves of different plants. Images taken in 2011, in the initial stage of the study (photos by J.A. González).

**Figure 7 biology-11-01566-f007:**
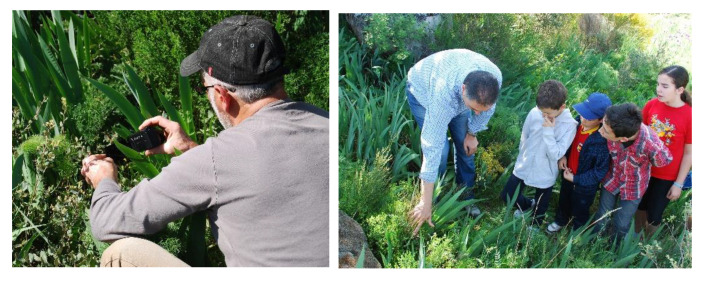
Different images of the educational activities implemented (photos by J.A. González).

## Data Availability

All data are included as part of the article.

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
