# Peer review of "Plant Conservation vs. Folk Traditions: The Case of Ophrys scolopax Cav. (Orchidaceae) in Central Western Spain"

_biology, 2022, doi:10.3390/biology11111566_

Round 1

Reviewer 1 Report

I have read this interesting and well-written paper about the importance of Ophrys scolopax in the the view och villagers in a specific area of Spain. The orchid plays some minor role in the customs and local lore and is therefore of importance for the village identity. It gives information how education practices (awareness campaigns) in the village have been of importance for the conservation of the species. The illustrations are good and the map informative. I think this paper can published as it is! 

Author Response

Thank you very much for your nice words about the acceptance of our article for publication in the leading journal “Biology” in the present form. We indeed believe that it highlights the need to introduce different educational activities in the field of biocultural heritage conservation.

Reviewer 2 Report

Thank you for your hard work in conserving biodiversity. It is impossible to succeed the conservation activities without the cooperation from local people.  However, because this journal is a scientific journal, the presentation, and the scope of paper should be focused on the scientific method and data rather than sociological or historical information. This information is interesting, but I feel like it is too much here, and the scientific data is obscure.

I my opinion, my paragraphs or quotes could be removed or summarised, e.g., 

- quote in line 147-160

- Figure 3

- Discussion of the vernacular names

Below are other opinions in my point of view

- The result in topic 3.1 should be presented in visual graphic

- I did not agree that the lack of knowledge in the children under 10, that you have discussed in paragraph 412, could be concluded that the TK transmission is broken down.

- The authors did not provide clear information about the TK of the bee orchid plant

- The information about the plant provided in line 571-595 is not a result, it should be part of the introduction.

- The authors did not clearly demonstrate the impact of education projects on the population or the conservation of the orchid

- What is the point of topic 3.2.2 for the global reader?

Author Response

Thank you for your hard work in conserving biodiversity. It is impossible to succeed the conservation activities without the cooperation from local people. However, because this journal is a scientific journal, the presentation, and the scope of paper should be focused on the scientific method and data rather than sociological or historical information. This information is interesting, but I feel like it is too much here, and the scientific data is obscure.

The work of our research group focuses on the conservation of biodiversity (mainly in the conservation of plant species), but also on the cultural aspects of the use of natural (floristic) resources… and the need to combine both fields of knowledge. We have sent this article to the journal “Biology” because we have been invited to participate in a special issue dedicated to Bio-Cultural Diversities Conservation, where we have believed from the beginning that our study-project has a place. Obviously, sociological or historical information has too much weight, we are aware, but it is necessary to make readers aware of the cultural background that threatens this plant.

Thank you very much for your comments and suggestions. In response to them, we have carried out a series of changes that we believe improve the final text of the article.

I my opinion, my paragraphs or quotes could be removed or summarised, e.g., 

- quote in line 147-160

OK… the texts of the ancient books have been removed.

- Figure 3

This figure is important to give a visual idea of ​​how the festival that is celebrated annually develops... and that coincides with the flowering period of the threatened orchid.

- Discussion of the vernacular names

OK… we have deleted this paragraph.

Below are other opinions in my point of view

- The result in topic 3.1 should be presented in visual graphic.

OK… in the subsection 3.1.5 (“Who collected it? (a robot portrait)”) data related to the collection of specimens by the people interviewed were represented in a chord diagram (information visualization). We think that this is not necessary in other points of “topic 3.1”.

- I did not agree that the lack of knowledge in the children under 10, that you have discussed in paragraph 412, could be concluded that the TK transmission is broken down.

OK… this sentence has been removed from the text. It is indeed very daring to make such a claim.

- The authors did not provide clear information about the TK of the bee orchid plant.

Traditional knowledge related, for example, to a material use of the bee orchid species does not exist. All the data recorded in relation to its collection by the inhabitants of Villarino de los Aires are mentioned in the Introduction and Results and Discussion sections.

- The information about the plant provided in line 571-595 is not a result, it should be part of the introduction.

This paragraph has been included at this point of the manuscript with the simple intention of explaining to the reader what concepts and data about the biology of the plant to be conserved we have explained to the inhabitants of the rural community involved in the project. As you can see, the second part of this paragraph relates its pollinators to the global threat situation suffered by insects in general.

- The authors did not clearly demonstrate the impact of education projects on the population or the conservation of the orchid /////// - What is the point of topic 3.2.2 for the global reader?

Subsections 3.2.2 and 3.2.3. they have been joined in a single sub-section entitled “Impact and Social Implications of the Project”. Likewise, a new final paragraph explaining the positive impact of the educational activities carried out on the population of the treated bee orchid species has been included in the text.

Reviewer 3 Report

The manuscript is very well prepared. I was very happy while reading. I got inspired.

Author Response

Thank you very much for your effort in reviewing our article. It is very gratifying to know that a reviewer like you has enjoyed reading our manuscript (we have always thought that this work should be pleasant in its presentation). Likewise, it is a full satisfaction to know that it has inspired you for your future research.

Round 2

Reviewer 2 Report

-